

# Design and fabrication of an electrostatic precipitator for infrared spectroscopy

Nikunj Dudani[1] and Satoshi Takahama[1]

[1]ENAC/IIE, Laboratory for Atmospheric Processes and Their Impacts (LAPI), École Polytechnique Fédérale de Lausanne (EPFL), 1015 Lausanne, Switzerland

**Correspondence:** Satoshi Takahama (satoshi.takahama@epfl.ch)

**Abstract.** Infrared (IR) spectroscopy is a direct measurement technique for chemical characterization of aerosols that can be applied without solvent extraction thermal treatment a priori. This technique has been used for chemical speciation, source apportionment, and detailed characterization of the complex organic fraction of atmospheric particles. Currently, most of the IR analysis is performed by transmission through porous membranes on which the particles are collected via filtration. The

membrane materials interfere with the IR spectra through scattering and absorption that not only makes extracting the chemical information of aerosol harder but also limits the lower extent of detection. An alternative IR measurement method that does not inherit such limitations aerosol is to collect the particles on an IR transparent material. We present an electrostatic precipitator design that enables such measurements by collection on a Zinc Selenide (ZnSe) crystal. Through numerical simulations and rapid prototyping with 3D-printing, we design and fabricate a device which is tested with poly-dispersed ammonium sulfate

particles to evaluate the quantitative chemical composition estimates against particle count reference. Furthermore, with an image analysis procedure and using variable aperture of the IR spectrometer, we analyze the radial mass distribution. The collector has high collection efficiency ($82\% \pm 8\%$) and linear response to mass loading ($R^2 > 0.94$) with a semi-uniform deposition. The method of design and fabrication is transferable to other applications and the current ESP collector can provide directions for further design improvements.

**Keywords**: Design, fabrication, collector, quantitative, IR, FT-IR, ESP.

## 1   Introduction

Mid-infrared (IR) spectroscopy is a non-destructive technique that can be used to directly probe the chemical signature of particles without solvent extraction or similar transformations for sample preparation. Absorbance signatures of major components of $PM_{2.5}$ (organic matter, elemental carbon, mineral dust, and inorganic salts) (Ivlev and Popova, 1973; Cunningham

et al., 1974; Bogard et al., 1982; Mcclenny et al., 1985; Pollard et al., 1990; Debus et al., 2021) and tracer species (Yazdani et al., 2021) can be found in the IR spectrum. To obtain an IR spectrum of the particles, they must first be isolated from the surrounding vapor as they also absorb IR and are present in comparable or greater concentrations as particle constituents.

Capability for quantification of chemical concentrations and source contributions have been demonstrated in transmission mode analysis on particles isolated on "optically thin" polytetrafluoroethylene (PTFE) membrane filters with the judicious



use of appropriate calibration standards and algorithms (Maria, 2003; Reff et al., 2007; Russell et al., 2009; Takahama et al., 2013; Ruthenburg et al., 2014; Takahama et al., 2019). However, this approach suffers from measurement uncertainties due to spectroscopic interferences from the substrate, as PTFE filters have strong absorption peaks in the fingerprint region and also scatters radiation (Mcclenny et al., 1985), which are partially handed by statistical algorithms for removal (Kuzmiakova et al., 2016; Yazdani et al., 2021). Reflectance spectra of particles collected on quartz or glass fiber filters or reflective foils vary in

magnitude of interferences but essentially require nonlinear calibration functions for quantitative analysis (Tsai and Kuo, 2006; Hopey et al., 2008; Parks et al., 2021).

One solution to this problem is to collect particles directly onto optical crystals transparent to IR for transmission mode analysis, but no method exists to achieve high collection efficiency on such substrates. Filtration is not amenable with conventional IR transparent crystals. Impaction has been used for collection onto such crystals in the past (Fischer, 1975; Allen et al.,

1994; Blando et al., 1998; Sax et al., 2005), but have material-dependent bounce off effects, size stratification, and undesirable deposition characteristics for spectroscopy (Marple, 2004; Blando et al., 2001; Virtanen et al., 2010). ESP is a versatile method and has low pressure drop unlike both the other methods, and is selected for this application.

ESP geometry for particle collection generally fall into two broad categories: a translationally symmetric design (linear ESP) (Harrick and Beckmann, 1974; Ofner et al., 2009) or a radially symmetric design (radial ESP) (Dixkens and Fissan, 1991; Fierz,

2007; Kala et al., 2012). In both systems, the electric field is perpendicular to the fluid flow that is parallel to the collection surface. Additionally, in most radial ESPs, the fluid enters the collection region through a tube perpendicular to the collection surface, before moving radially outwards. ESPs can also be categorized by single-stage or two-stage configurations, depending on whether the charging and collection is handled with the same or different set of electrodes — the former is simpler in design, while the latter permits better control in deposition. However, existing designs across these categories either require

high electric field strengths for operation that can cause chemical modification of the aerosol, or have high size-segregation in deposition.

We therefore introduce a new two-stage radial ESP for collection of particles on IR transparent crystals, and report on its capability for quantitative IR analysis. A charger with low chemical interference is selected from the existing literature, while a new radial collector is designed to permit high collection efficiency and mass throughput. The choice of charger is germaine

since the strong ionization in the chargers result in formation of reactive molecules such as $O_2^+$, $O^+$, $N_2^+$, $N^+$, $NO^+$, and $H_3O^+$ (Volckens and Leith, 2002; Arnold et al., 1997), leading to changes in particle composition through ozone reactions and gas-to-particle conversion (Volckens and Leith, 2002). Numerous chargers have been developed and use either through direct corona discharge to charge the particles (Hewitt, 1957; Liu and Pui, 1975; Biskos et al., 2005; Whitby, 1961; Tsai et al., 2010), or indirect corona discharge where charged ions in gas flow are generally mixed with the particle flow separately (Medved et al.,

2000; Marquard et al., 2006; Kimoto et al., 2010). Indirect chargers have higher charge levels but leads to diluted particle concentrations after mixing, which reduces mass throughput. Direct or indirect photoelectric discharge (Burtscher et al., 1982; Grob et al., 2014; Nishida et al., 2018; Shimada et al., 1999) — especially with UV photoionization (Hontañón and Kruis, 2008; Grob et al., 2013) — leads to less chemical artifacts, but generally have strong dependence on the conductivity of the particle (which varies by composition). Two different unipolar chargers with minimal ozone generation suitable for chemical





sampling have been proposed: one using carbon fiber ionizer technology Han et al. (2008) and another using a wire-to-wire
      configuration of metal electrodes Han et al. (2017). We adapt the latter for this work.

          The radial collector design, numerical simulation, and novel methods for fabrication using 3D printing and post-treatment
      (together with the rest of the ESP) are described in the rest of the manuscript. The remaining portion of the manuscript describe
      the methods and results for characterization of collected mass and its relationship to apparent IR absorbance.

## 2    Method

      In this section, we describe the design objectives and constraints (Section 2.1), numerical simulations for virtual performance
      characterization (Section 2.2), and 3D printing (Section 2.3) with acrylonitrile butadiene styrene (ABS). The fabricated de-
      vice and an aerosol flow system (Section 2.4) was used to collect ammonium sulfate particles on a 25.4 mm diameter ZnSe
      crystal, which has desirable chemical, electrical, and optical properties for this application. The IR spectra of this crystal is
acquired before and after sampling (Section 2.5). The spatial deposition pattern on this crystal was also characterized using
      optical microscopy, electron microscopy and IR analysis (Section 2.6). All of these measurements are combined to evaluate
      the quantitative response of the IR measurement with particle loading (Section 2.7).

### 2.1    Design of a radial ESP

      A two-stage radially symmetric (radial) ESP was selected for geometry and flow design. Separating the charging and collection
permits greater control over the conditions for charging and collection within each stage. The radial collector configuration does
      not have flow and collection directionality that is inherent in a linear ESP, which could theoretically result in some directional
      dependence in the particle deposition based on particle size and flow rate; and the radial symmetry in deposition profile is
      consistent with the transmission IR beam. The resulting design blueprint has a particle charger connected to an ESP where the
      flow enters perpendicular to the collection surface and moves radially outward and subsequently collected, illustrated in the
device schematic (Figure 1a).

          One of the critical requirements in the design is to limit the high electric field strength and voltage in the vicinity of the
      particles, mainly because regions of high electric field facilitate ion production that modifies the chemical composition of the
      particles (Ofner et al., 2009). For the charging stage, another application where ozone production is an undeniable consideration
      is personal bio-aerosol sampling and the PEBS's wire-to-wire electrode arrangement in the charger (Han et al., 2017) was
specifically designed to maintain a low ozone concentration ($< 10$ ppb). In this work, we employ a similar wire-to-wire charger
      and design the collector stage with the following desired features (1) Low electric field strength (lower than 1 kV/mm, with a
      factor of safety 3 over the theoretical breakdown field of air at 3 kV/mm); (2) Low electrode voltage (lower than 10 kV); (3)
      High collection efficiency (greater than 70%); (4) High flow rate (for example, a 1.7 LPM ($2.8 \times 10^{-5}$ m$^3$/s) flow rate would
      result in a 1 µg total mass collection in 1 hour for an ambient concentration of 10 µg/m$^3$).

90        In the collector, the combination of an electric field limit to 1 kV/mm and a voltage limit of 5 kV imposed a minimum
      electrode separation distance of 5 mm. For conditions where inertial effects and diffusion effects can be neglected, the collection



in the radial ESP is a result of the trade-off between the drag force on the particle parallel to the collection surface ($r$-direction) and the electrostatic force into the surface ($-z$-direction). For a given flow conditions, a stronger electric field will result in a more efficient particle collection making closest 5 mm separation the most desirable as any larger separation distance would either require that the voltage be higher or $E_0$ be lower than 1 kV/mm.

Based on these design constraints, prototypes were developed using 3D-CAD software (Section S1 Figures S2a, b, c), simulated for particle collection (Section 2.2), and assembled after 3D-printing and post-treating (Section2.3 and Section S1 Figure S2d). The charger (modeled after Han et al. (2017)) was 3D printed using ABS and had the same dimensions of a 25.4 mm diameter tube, with the circular ground electrode wire made from 0.5 mm diameter stainless steel wire, and the ionizer was a 0.075 mm diameter Tungsten wire, 25.4 mm in length sitting symmetrically about the ground electrode while held rigidly in place at the ends through two thin glass tubes (1 mm outer diameter) that were connected to the ABS housing. The total length of the flow channel was 60 mm, and had a further gradually expanding and contracting part at either ends (40 mm in length and having 10 mm diameter at the other ends, where stainless steel tubes were used to connect the charger to the inlet and the collector).

## 2.2 Numerical simulations

Particle trajectories were numerically simulated in an electrostatic and flow field using COMSOL Multiphysics software. 2D-axisymmetric simulations allow much faster simulations and the fluid flow was simulated using laminar flow physics. The simulation was made with dimensions and materials replicating the actual fabricated device. A radial ESP collector (made of ABS) was simulated with a tubular inlet facing the substrate (ZnSe crystal) resting on the ground electrode, while the high voltage electrodes were places near the outlet of the tube and at a fixed distance above the collection surface (Section S1 Figure S1a) using an extremely fine physics-controlled mesh. The electrostatic field (Section S1 Figure S1b) and the fluid flow field (Section S1 Figure S1c) were simulated using a stationary solver with $10^{-3}$ relative tolerance for convergence. Despite large scale eddies in the laminar flow simulations, no turbulent simulation was required mainly because mesh-refinement analysis yielded identical laminar flow results (for example, changing the mesh to a finer grid resulted in similar flow fields). This was confirmed using mesh-refinement in a 3D simulation of the device design with laminar flow field. The in-variance to mesh-refinement suggests that the obtained calculations were indeed resolved with laminar physics alone and did not represent a scrupulous flow field.

Time-dependent particle trajectories were simulated in the two stationary fields (Section S1 Figure S1d) with convergence at $10^{-5}$ relative tolerance. No coupling of the perturbations of the particles on the stationary fields is employed as the particles are very small to cause substantial change. 1000 charged particles were released uniformly spaced at the top of the inlet tube at $t = 0$ till it collected on the surface, collided with another surface or ran-off. The charge levels on the particle was assumed to be proportional to particle diameter (as assumed for diffusion charging) with around one elementary charge for every 20 nm diameter (Biskos et al., 2005). A couple of additional charge values around the linear value were also simulated for each particle size as the charge can be expected to be higher on larger particles if particle charge is a combined effect of field and diffusion charging (Marquard, 2007). The particle diameter $D_p$ was manually adjusted for the slip corrected factor





$C_c = 1 + \text{Kn}\left[1.142 + 0.558\exp\left(-0.999/\text{Kn}\right)\right]$ (Allen and Raabe, 1985), where Kn is the Knudsen number, in the drag force calculations for the particle simulation.

An iterative simulation process was used to explore geometries and operating parameters to find a configuration permitting high flowrates (for high throughput), and long residence time of particles between the electrodes (to improve collection efficiency) within electric field constraints described above. At the end of each simulation, the position of the particles on the crystal are used to calculate the histogram of the relative frequency at different radial positions (Figure 1b) as it represents the spatial distribution. Collection efficiency was calculated using the number of particles out of the 1000 that were collected on the surface (i.e till $r < 12.7$ mm) – measured through the cumulative frequency from the histogram in Figure 1b. A device with inlet radius of 10 mm and flow rate of 2.1 LPM, and $E = 1$ kV/mm was estimated to have a collection efficiency of 75% (for 200 nm particles with 8 elementary charges) for $D_p$ of 100, 200, 400, 600, and 800 nm. An additional increase in collection efficiency of 5% was estimated for incorporating a protrusion that extended the inlet tube closer to the collection surface while keeping the electrodes at a farther distance (Section S1 Figure S1a). We satisficed with this design for fabrication.

The resulting device is a radial ESP with the particles entering through a 10 mm diameter tube, perpendicular to a 25.4 mm diameter and 5 mm thick ZnSe crystal, which sits on the ground electrode, and the top electrode is positioned at 5 mm above the top crystal surface and at a diameter of 36 mm around the crystal (such that the minimum separation between the crystal surface and the electrode is 5 mm). Furthermore, the entry tube is extended by 4 mm (referred as "protrusion") and brought closer to have the exit of the tube 1 mm above the crystal. The electrodes are made using Ag or Cu materials, and the other parts of the device are 3D printed with ABS (including the tube extension). The device is operated to maintain an $E_0 = 1$ kV/mm, voltage difference of $V_0 = 5$ kV, and operating at a flow rate $Q = 2.1$ LPM.

This work was completed prior to the publication of an analytical radial electrostatic collector model by Preger et al. (2020). Their model estimates collection spot size as a function of flowrate and electric field strength in a radial collector with parallel plates and small inlet radii, which could now be used in selection of collector plate or operating parameters for devices which follow these geometric constraints.

## 2.3 Fabrication via 3D printing

We discuss three considerations for fabrication: material selection, printing protocol, and post-printing assembly and treatment. For rapid-protyping there are limited materials that can be 3D-printed reliably: nylon, ABS and polylactic acid (PLA); these materials have tradeoffs in print reliability, durability, chemical interferences, and electrical properties. PLA is not stable to heat, less durable, and has known outgassing issues, limiting its application for aerosol chemical analysis. Nylon is a stronger material than ABS but lies higher in the triboelectric series, making it prone to more electrostatic losses of particles near the surface. Moreover, in our observations the main source of leaks in the initial prototypes was through the in-layer-space between each print layer. The spacing was much higher in nylon as it can absorb moisture during and after printing and results in larger printing defects and layer-separation. The additional strength of nylon comes with cost of being more brittle and harder to seal, leaving ABS as our choice of material for this application. The strength of ABS is sufficient for inherently low pressure drop





ESP applications as it has a yield strength of 25 MPa, which is much higher than the expected hoop stress of 1.7 MPa acting on a 50 mm internal diameter cylinder design with a thin 2 mm ABS body operated with an extreme 1 atm pressure.

The 3D Systems CubePro 3D printer was used to fabricate all printed parts. Warping, cracking, curling and stringing was reduced with proper printing speed, temperature and layer spacing adjustment. The print bed was regularly leveled to better adjust to the filament feed rate, leading to smoother printed objects. The feed-rate and temperature was self-controlled by the CubePro 3D printer and the print chamber was kept heated generally up to 55 °C (failure in the chamber heating results in poor print quality due to spatially uneven cooling). Using a pre-heated print chamber along with a water soluble glue that was applied on the print surface and dried completely before starting each print job substantially improved bed-adhesion, which can be problematic in longer ABS printing jobs. Finally, the ESP was printed with as few parts as possible to have higher structural stability and avoid forced turbulence because of material discontinuity at the joints. This strategy required fabrication of parts with complex overhangs. Using filler materials with dual head printers is a possible solution, but has limited application for larger parts — as the printing volume and time scales cubically to the size on top of which printing the entire hollow part with a filler substantially increases the chances of print failure. We instead pursued an alternate strategy of printing the parts tilted at 45°. The parts which have an angle of 90°or lower ranged within ±45°with respect to the bed and hence were all inter-supported, and further strengthened by using simple line or point supports on the surfaces. Increasing the size of the base on the print bed promoted bed-adhesion, and further use of side supports enabled reliable printing of surfaces supported at the largest theoretical angle (45°).

Further post-assembly and treatment strategies were necessary to reduce gas leak through 1) areas where two parts join and 2) the layer spacing in the print surface. The junctions where two parts assemble were sealed by extending the joining surfaces outward to act as larger flanges, and were sealed using an o-ring placed in a groove designed for the static axial (face) assembly where the pressure is lower on the interior. The flanges and the greased (Using GE Bayer Silicones Baysilon grease) o-ring were assembled together using multiple screws along the circumference, using latches, magnets, or clips. In our observation, even using a calculated number of screws based on the theoretical bite angle sometimes resulted in leaks. A reliable solution was to use a series of magnets arranged in a circumferential Halbach array that directs the field in the axial direction. Two such arrays of 16 magnets were designed where each array was housed in a single 3D printed part shaped as a leaf spring to redirect the axial force evenly over the flange. The combined force applied was around 1 kN and was sufficient to eliminate all leaks. Another practical advantage of using this design was that despite the very strong force, opening the flange was possible by rotating one of the arrays such that opposite polarities align - resulting in an easy disassembly and reassembly, useful for changing the ZnSe crystal between experiments.

The inter-layer spacing in the 3D prints was sealed along with smoothening the rough 3D-printed surface by manually rubbing the outside surface with acetone while pressing down on the surface until no spots were left. Since ABS dissolves in acetone, surfaces are sometimes smoothed by suspending an ABS part in a chamber above an acetone vapor bath. In our observation, the method resulted in overly softened parts or unsealed portions numerous times due to the sensitivity to the bath time and the acetone amount. Therefore, the surface and joint sealing procedure was repeated until the assembled device could hold vacuum at levels of −10 kPa for over 5 minutes. To locate sources of leaks, positive pressure was created using





a compressed air flow through the assembly (static pressure in closed system to avoid a pressure buildup inside 3D printed
parts, which is a safety concern) and a soap solution was applied over the entire printed surface and joint areas. Leak locations
were identified by the resulting bubbles and then sealed. Sealing larger gaps in nylon parts required a coat of epoxy solution
thinned using acetone for easier application, which required longer preparation times and higher chances of smaller holes
remaining after treatment because of the higher surface tension of the epoxy solution, further enforcing advantages of ABS for
prototyping.

## 2.4 Aerosol flow system

Laboratory-generated ammonium sulphate ($(NH_4)_2SO_4$) particles were used for ESP performance evaluation. Two flow system
setups were used: one to characterize the generated and collected particle size distribution using a scanning mobility particle
sizer (SMPS) (Figure 2a), and another to measure the total particle counts collected at higher time resolution over the course
of each experiment with a condensation particle counter (CPC) (Figure 2b).

Ammonium sulphate particles were continuously nebulized from freshly prepared 5 g/L concentration solutions in milli-Q
water with TSI, Inc., 3076 atomizer and dried using silica gel denuder (machined in-house). The particles were passed through
a cyclone to remove supermicron particles and the concentration was adjusted with a dilution system before introducing the
particles into the ESP. Different bypass lines (red-dotted flow lines) were setup to measure the size-distribution or particle
count at different flow-line points. A mass flow controller (MFC, MKS) with a vacuum pump (GAST) was used to maintain
the desired ESP flow (2.1 LPM).

The charger was operated at a 6 kV voltage on the Tungsten wire, and the ESP was operated at 5 kV voltage difference
between the top electrode and the top surface of the ZnSe crystal. We used a positively biased ground electrode in the ESP (at
1 kV) to avoid potential negative voltage in any part of the assembly. Moreover, as the ZnSe crystal is not perfectly conducting,
we observed that the top surface of the ZnSe crystal was around 1 kV voltage higher than the ground electrode voltage, resulting
in a top electrode voltage of 7 kV for maintaining the 5 kV voltage difference. The entire assembly was operated at 2.1 LPM
flow rate, maintained using two pumps - one connected through the CPC and the other through a mass flow controller.

Particle size distributions $n^\# = dN^\#/d\log D_\mathrm{p}$ between 17.5 nm to 982 nm were measured with a scanning phase of 210 s
and a down-sampling time of 30 s with an aerosol flow rate of 0.7 LPM and a sheath flow rate of 2 LPM. Size measurements
were obtained with 6, 20 and 33 scans at three points of interest: at the inlet of the ESP ($n^\#_\mathrm{in}$), the outlet of the charger
($n^\#_\mathrm{ch,out}$) and the outlet of the collector ($n^\#_\mathrm{col,out}$). Additionally, $n^\#_\mathrm{ch,out}$ and $n^\#_\mathrm{in}$ were measured again at the end of the experiment,
with 6 and 13 scans respectively. The mean volume of collected particles $\overline{V}_\mathrm{p}$ was computed from the collected distribution
$\left(n^\#_\mathrm{ch,out} - n^\#_\mathrm{col,out}\right)Qt$ with flowrate $Q$ and collection interval $t$.

The transient raw particle count experiment (Figure 2b) was conducted for different collection intervals, $t$ (from 5 minutes
to 3 hrs) and adjusting the particle concentration $N^\#_\mathrm{ch,out}$ to around $9000 - 10000$ particles/cm$^3$ i.e. below the operating limit
of the CPC. The flow line is switched from measuring $N^\#_\mathrm{ch,out}$ from the bypass to $N^\#_\mathrm{col,out}$ in the main line at a resolution of
one measurement per second. Because the particle number may drift during the duration of the experiment, the average and
deviation in number concentrations over this period ($\bar{N}^\#_\mathrm{ch,out}$ and $\delta\bar{N}^\#_\mathrm{ch,out}$, respectively) are characterized for calculation. The



corresponding difference in the concentrations between charger and collector outlet is used to estimate the number of particles collected ($N^*$) and its error ($\delta N^*$).

$$N^* = \left( N^{\#}_{\text{ch,out}} - N^{\#}_{\text{col,out}} \right) Qt, \ \delta N^* = \delta N^{\#}_{\text{ch,out}} Qt \tag{1}$$

$N^*$ and $\overline{V}_{\text{p}}$ are used to calculate the collected mass (Section 2.7).

## 2.5 Infrared spectroscopy

The IR absorbance spectra of the particle loaded and clean ZnSe crystal were measured with the Vertex 80 with Deuterated Lanthanum $\alpha$ Alanine doped TriGlycine Sulphate (DLaTGS) detector. Each spectrum was measured after purging the measurement chamber for 3 minutes with dry, compressed air (from Bruker gas generator) after introducing the sample, and was measured with an average of 64 scans over $4000 - 400$ cm$^{-1}$ with 4 cm$^{-1}$ resolution. The purged chamber was used as the background for measurement of the sample and clean crystal spectra. The default aperture (Jacquinot stop) setting of 6 mm diameter was used, except for a set of experiments to study the spatial distribution of deposited particles (VAIRS, Section 2.6). The beam spot is approximately $80\%$ greater in diameter than the aperature setting (Bruker rep., personal communication, 2018). Two identical ZnSe collection crystals (WG71050 from Thorlabs Inc.) were used for all the experiments. The crystal was cleaned between each loading and IR measurement by alternating between rubbing with a clean wipe (Kimberly Clark 7552) dampened with iso-propyl-alcohol (IPA, $=99.8\%$ (GC grade) Sigma Aldrich) and another with milli-Q water.

The raw IR spectra is processed to correct small deviations in the baseline which can arise from changes in background conditions in the IR chamber. In this work, $(NH_4)_2SO_4$ was the only test material and a method that automatically corrects the baseline is employed. First, the absorbance spectra of the cleaned collection crystal is subtracted from the particle-loaded absorption spectra. The slope of absorbance values is used in combination with the moving mean of absorbance to identify non-absorbing baseline points (Section S2). A smoothing spline was fitted to interpolate the baseline through absorption regions and then subtracted from the original spectrum.

## 2.6 Spatial profiling

The spatial profile is primarily assessed using optical image analysis, and supported by electron microscopy and a method we refer to as variable aperature infrared spectroscopy (VAIRS) in this work. A top-view image of the ZnSe crystal was taken with an optical camera after each deposition experiment. Using a dark background to contrast the bright ammonium sulfate particles, we took an image of the particle loaded crystal under diffused light using a digital camera (Firefly DE300, focal dept of 35mm with 2048 ×1536 pixel resolution) kept at a fixed height of 30 cm. The image was analyzed by converting to gray scale (0 to 255 pixel intensity), and finding the radial change in pixel intensity from the analyzed center and periphery of the crystal.

Scanning electron microscopy (SEM) analysis was performed to qualitatively assess the particle distribution at different radial positions using the FEI Teneo microscope. Secondary electrons were detected using the Everhart-Thornley detector





detector while using a 5 kV voltage and low 50 pA current to prevent charge build up on the particles. The crystal was
introduced in the SEM chamber without any conductive coating and images were taken from a working distance of 6 mm.

For VAIRS analysis, IR absorbance intensities for different apertures (0.25 to 8 mm) were used to compute the surface mass
distribution. By varying the aperture setting, and consequently the beam spot size on the surface (which is around $80\%$ larger
than the aperture), the observed change in absorbance intensity of the peak near 1410 cm$^{-1}$ was representative of the areal mass
density of the deposition under the beam. From the difference in intensities among successive beam areas, the mass per unit
annulus area was estimated.

## 2.7 Quantitative evaluation

The quantitative analysis of spectra outlined in this work is enabled through the condition that small particles are spaced far
apart (weak scattering approximation) on the collecting crystal. Further details for the following statements are provided in
Section S4 and S5. The absorbance contribution from particles is obtained from subtracting the absorbance of the clean crystal
from that of the sample. The transmittance $T$ of the sample used for calculation of absorbance is obtained by ratioing the
sample spectrum to single-beam background of the purged chamber, and transmittance $T_\mathrm{c}$ of the clean crystal is obtained
prior to the experiments by ratioing a clean ZnSe crystal to the purged chamber background. Neglecting scattering interactions
between particle and crystal and among particles, the absorbance $A$ is related to the mass of deposited particles per unit area
of a disc, $m_\mathrm{a}^*(r_\mathrm{b})$, inscribed by the beam radius $r_\mathrm{b}$, substance density $\rho$, and the decadic volume attenuation coefficient $\alpha_{10,\mathrm{v}}$
(the Naperian volume attenuation coefficient divided by $\ln 10$) (Adamson, 1979; Bohren and Huffman, 1983; Hapke, 2012):

$$A(\tilde{\nu}) = -\log_{10} T(\tilde{\nu}) + \log_{10} T_\mathrm{c}(\tilde{\nu}) = \alpha_{10,\mathrm{v}}(\tilde{\nu})\frac{m_\mathrm{a}^*(r_\mathrm{b})}{\rho} \ . \tag{2}$$

$\alpha_{10,\mathrm{v}}/\rho$ is the decadic form of the mass attenuation coefficient, which can be related to the molar attenuation coefficient
previously used to characterize IR response (Allen et al., 1994; Maria, 2003), and also the molecular absorption cross section
used in other applications. Alternatively, the areal mass density $m_\mathrm{a}^*(r_\mathrm{b})/\rho$ can be conceptualized as the equivalent thickness of
the sample in the form of a void-free film (without coherent interferences).

The mean areal density over the entire crystal $m_\mathrm{a}^*(R)/\rho$ is calculated from the total deposited volume $m^*/\rho$, estimated from
the product of the SMPS and CPC measurement outputs:

$$\frac{m_\mathrm{a}^*(R)}{\rho} = \frac{m^*}{\rho\pi R^2}, \text{ where } \frac{m^*}{\rho} = \overline{V}_\mathrm{p} N^* \ . \tag{3}$$

$N^*$ is the cumulative number of particles deposited from the CPC (Eq. 1 and Section 3.1), $\overline{V}_\mathrm{p}$ is the mean volume obtained
from the measured size distribution from the SMPS (Section 3.1), and $\rho$ is the mass density of the substance (ammonium
sulfate). The mean areal density under the beam is calculated through an adjustment factor $\phi_R(r_\mathrm{b})$:

$$m_\mathrm{a}^*(r_\mathrm{b}) = \phi_R(r_\mathrm{b})m_\mathrm{a}^*(R) \ . \tag{4}$$





The value of this adjustment factor is calculated from the scattered intensity $I^*$ (Section 3.2) integrated over various radii, which is interpreted as being proportional to mass loading:

$$\phi_r(r_\mathrm{b}) = \frac{m_\mathrm{a}^*(r_\mathrm{b})}{m_\mathrm{a}^*(R)} \approx \frac{I^*(r_\mathrm{b})/r_\mathrm{b}^2}{I^*(R)/R^2} \ . \tag{5}$$

From measurement, $\alpha_{10,\mathrm{v}}(\tilde{\nu})$ is estimated from individual experiments by dividing the apparent absorbance $A(\tilde{\nu})$ by $m_\mathrm{a}^*(r_\mathrm{b})/\rho$, or collectively determined by calibration for specific wavenumbers at peak apexes. These values are compared against theoretical values obtained from previously reported refractive indices of ammonium sulfate. Given the size range of particles used in our study (Rayleigh regime), absorption is the dominant process leading to attenuation of IR radiation by particles. In the electrostatics approximation (van de Hulst, 1957), the volume attenuation coefficient of such small particles is related to the vacuum wavenumber $\tilde{\nu}$ and complex refractive index $\tilde{n}$ of the substance comprising the particles (Bohren and Huffman, 1983):

$$\alpha_{10,\mathrm{v}}(\tilde{\nu}) = \frac{6\pi\tilde{\nu}}{\ln 10} \,\mathrm{Im}\left\{ \frac{\tilde{n}^2(\tilde{\nu}) - 1}{\tilde{n}^2(\tilde{\nu}) + 2} \right\} \ . \tag{6}$$

In contrast, the decadic linear absorption coefficient $\alpha_{10}$ that would be used in place of $\alpha_{10,\mathrm{v}}$ in Eq. 2 for a homogeneous medium follows a different relation with the refractive index and is also used as a point of comparison:

$$\alpha_{10}(\tilde{\nu}) = \frac{4\pi\tilde{\nu}}{\ln 10} \,\mathrm{Im}\{\tilde{n}(\tilde{\nu})\} \ . \tag{7}$$

For a collection of tenuous particles (Rayleigh-Gans-Debye approximation), the volume absorption coefficient and linear absorption coefficient are related in the weak absorption limit as $\alpha_{10,\mathrm{v}} \approx \alpha_{10}$ (Bohren and Huffman, 1983), but this condition does not strictly apply here due to the sharp contrast in the refractive index of ammonium sulfate and that of the surrounding medium (air). $\alpha_{10,\mathrm{v}}$ is evaluated in Section 3.3. The deviation from 1 between the ratio of $\alpha_{10,\mathrm{v}}$ to $\alpha_{10}$ is significant especially near the absorption peaks, calculated from reference spectra of Ammonium Sulfate (Earle et al., 2006) (Section S3 Figure S5a).

## 3 Results and discussions

We discuss the generated and collected particle size distributions (Section 3.1) and spatial deposition profiles (Section 3.2) used to estimate the mass deposited within the probing area of the IR beam, and then evaluate how spectral absorbance relates to this estimated mass (Section 3.3).

### 3.1 Particle size distributions and collection efficiency

The measured average particle size distributions at the charger inlet ($\bar{n}_\mathrm{in}^{\#}$), charger outlet ($\bar{n}_\mathrm{ch,out}^{\#}$) and collector outlet ($n_\mathrm{col,out}^{\#}$) according to the described method (Section 2.4) shows a charger penetration of $78\% \pm 6\%$ (over the particle size range $D_p <$ 982 nm); and a collection efficiency of $82\% \pm 8\%$ on the fraction of particles at the charger outlet (Figure 3a). The charger efficiency was in accordance with the ones reported in the charger design (between 75% and 90%) at the operating voltage of 6 kV but measured only for higher flow rates (Han et al., 2017). The mass mean diameter of the collector fraction was 207 nm and an additional 5% mass was estimated to have been present in the system, using a log-normal fit extension of the measured


volume distribution (which is limited till 982 nm). The log-normal distribution fit was used to obtain a tail that extended further from 982 nm and was scaled to match the volume distribution at the last bin (982 nm).

The Figure 3b shows corresponding normalized number size distribution, where the charger inlet distribution at both the
start and at the end of the experiment are included to provide an estimate of expected change in the distribution over the long sampling experiment. The collector did collect some larger particles preferentially but qualitatively there was no large shift in size distribution because of the charger or the collector. Contrary to the observed nominally higher collection for larger particles, COMSOL simulations showed a nominally decreasing collection efficiency with increasing particle size (as the electrical mobility was decreasing) for a charge level proportional to the diameter. The discrepancy is mostly because of larger
fraction of larger particles getting charged vs. un-charged in the charger. Numerical simulations had a similar $80\%$ collection efficiency for $D_\mathrm{p}$ = 100, 200, 400, 600, and 800 nm with 4, 8, 20, 36, and 48 elementary charges respectively. Qualitatively, the particle distribution imaged by the SEM (Section S1 Figure S3a, b, c) is similar at the three positions and support the idea that there is lower size segregation in collection. Particles were spatially separated, as expected below 5–20 µg/cm$^2$ (Casuccio et al., 2004), and supports the independent scattering assumption (Drolen and Tien, 1987) over the range of areal mass densities in
our experiment ($< 3.15$ µg/cm$^2$).

## 3.2   Spatial mass distribution

The three methods of evaluating the spatial profile (Section 2.6) show that the deposition is semi-uniform. The radial dependence of pixel intensity computed from optical images (Figure 4a) show a general trend for prominent mound near the point where the device inlet tube extrudes towards the surface ($r = 5$ mm), as predicted by simulation (Figure 1b). The presence of
the mound is supported by qualitative SEM image analysis which has nominally different particle densities for the different images at different radial positions (1, 6 and 9 mm) (Section S1 Figure S3a, b, c), despite having low qualitative variation in the size distribution. However, there are some observed variations in the relative height of the mound and whether the deposited mass is increasing or decreasing with radial distance. The VAIRS analysis (Section 2.6) of the IR absorbance spectra with different aperture sizes, also closely follows the image analysis profile (Figure 4), though it is much coarser because of limited
number of aperture points. These variations across experiments, especially the relative mound intensity, likely results from small perturbations in the vertical position of the crystal placed in the collector disc housing (Section S1 Figure S2a)

The scattered integrated over the entire crystal $I^*(R)$ for each experiment scales with the total mass areal density $m_\mathrm{a}^*$ calculated using image analysis (Section S3 Figure S7a) and supports its use for resolving the spatial distribution of deposited mass (albeit making use of variations in intensity over a single experiment). The estimated density scaling factor $\phi_R(r_\mathrm{b})$ for a
beam radius of $5.4$ mm corresponding to the aperture of 6 mm diameter is ratio of $I^*(r_\mathrm{b})/\pi r_b^2$ to $I^*(R)/\pi R^2$ (Figure 4b), and its deviation from unity (1:1 line in the Figure) is mostly $\pm 10\%$ (with two points at +20% and +25%). $\phi_R(r_\mathrm{b})$ is systematically higher for the intensity profiles that increases sharply with radius and lower for profiles which have decreasing radial intensities (Figure 4a). The non negligible deviation in $\phi_R(r_\mathrm{b})$ in conjunction with it conserving some detail about the distribution profile, we employ the density scaling factor to estimate the mass, $m_\mathrm{a}^*$, in this work (Section 3.3). A concurrent analysis is done in
Supplementary material without the image analysis correction i.e. with a value of $\phi_R(r_\mathrm{b}) = 1$ for all samples.





### 3.3 Absorbance with mass loading

We first examine features of the measured volume attenuation coefficient (effectively, its absorption profile) over the entire range of scanned wavenumbers. Then, we examine the magnitude of the volume attenuation coefficient at two of the major peaks. These quantities are compared against reference values derived from previously reported refractive indices of ammonium
sulfate obtained by different methods.

     The experimentally-determined volume attenuation coefficient are shown in Figure 5a. The absorbance peak locations and their magnitudes are generally consistent, indicating that the method of measurement and baseline correction does not introduce major chemical or spectroscopic artifacts. Some absorption peaks for aliphatic C-H (2800 - 3150 cm$^{-1}$) is visible, likely due to impurities on the crystal originating from the o-ring sealing grease in the crystal housing during manual removal of the sample,
or condensed vapors on optical components, or impurities in the solvent used for cleaning the crystal (resulting in negative peaks). The o-ring sealing grease most likely contaminates each experiment as most silicone greases use polydimethylsiloxane (PDMS) and absorption bands for both Si-CH$_3$ (800 cm$^{-1}$ and 1260 cm$^{-1}$) are visible for all experiments. PDMS IR spectra (Myers et al., 2001) also has a Si-O-Si band (1130 - 1100 cm$^{-1}$) that can potentially interfere with the SO$_4^{2-}$ absorption peak near 1090 cm$^{-1}$, but does not appear to materially affect SO$_4^{2-}$ quantification (Section S3 Figure S5a and S6b) and is therefore
not further considered in the analysis. The spectra does not appear to contain peaks related to nitrates or other additional artifacts that may be caused by spark discharge and formation of reactive molecules near regions of high electrical potential in the charger or collector. (Such peaks in initial stages of ESP-development were observed where needle electrodes and high voltages were used for charging and collection in a single stage design.) Moreover, no physical heating of the ZnSe crystal was observed after each experiment, which also suggests the absence of sparking and substantial production of reactive gases.
Overlayed on Figure 5a are $\alpha_{10,v}$ and $\alpha_{10}$ calculated from refractive indices measured by (Earle et al., 2006) using ammonium sulfate particles suspended in nitrogen gas in an aerosol flow tube. In our experiments, we observed a consistent peak positions of $\nu_3$(SO$_4^{2-}$) = 1090 cm$^{-1}$ ($\pm 0.2\%$) and $\nu_4$(NH$_4^+$) = 1415 cm$^{-1}$ ($\pm 0.08\%$), which are consistent with reported values for homogeneous samples (Toon et al., 1976) [$\nu_3$(SO$_4^{2-}$) = 1090 cm$^{-1}$, $\nu_4$(NH$_4^+$) = 1415 cm$^{-1}$] and (Torrie et al., 1972) [$\nu_3$(SO$_4^{2-}$) = 1093 cm$^{-1}$, $\nu_4$(NH$_4^+$) = 1417 cm$^{-1}$] — though the lack of wavenumber resolution in the older measurements
made on dispersive spectrometers prevent a more precise comparison. Blue shifting of peaks on the order of $\sim$10 cm$^{-1}$ can be expected for small particles (Bohren and Huffman, 1983; Maidment et al., 2018), with asphericity and increasing particle size reducing the extent of this shift (Mishchenko, 1990; Segal-Rosenheimer et al., 2009). Blue-shifts have been reported in extinction spectra of ammonium sulfate in aerosol flow tubes (Weis and Ewing, 1996; Earle et al., 2006; Segal-Rosenheimer et al., 2009; Laskina et al., 2014), though observations are often below that predicted by Mie theory. Such shifts are not observed in
our experiments, though the extent of peak shifts, spectral profiles, and the underlying refractive indices vary among various studies (Laskina et al., 2014; Johnson et al., 2020; Myers et al., 2020) and are attributed to some extent on differences arising from sample morphology.

     Figure 5b shows the absorbance against the effective deposition thickness for the $\nu_3$(SO$_4^{2-}$) peak near 1110 cm$^{-1}$ and $\nu_4$(NH$_4^-$) peak near 1410 cm$^{-1}$, the slope of which can be compared to $\alpha_{10,v}$ and $\alpha_{10}$ calculated from the refractive indices of





(Earle et al., 2006). The reference values are calculated for their respective peaks which vary by a few wavenumbers from our experiments, as mentioned previously. The experimental absorbance and effective deposition thickness estimates are strongly correlated ($R^2$ of 0.95 and 0.94, respectively). While weak bands are typically recommended for quantification of analytes of inorganic substances (Mayo et al., 2004) due to possibility for signal saturation, for low loadings (low optical depths) for which single scattering approximation applies, this work confirms that even the strongest absorbance band of $\nu_3(SO_4^{2-})$ at 1110 cm$^{-1}$

exhibits high linearity and can also be for quantification. The slope values of $1.32\times10^4$ cm$^{-1}$ (95% confidence interval (CI) of $[1.11, 1.52]\times10^4$ cm$^{-1}$) for the peak near 1110 cm$^{-1}$ and $5.89\times10^3$ cm$^{-1}$ (95% CI of $[4.87, 6.93]\times10^3$ cm$^{-1}$) for the peak near 1410 cm$^{-1}$ vary from the reference values within 20% and -8%, which is within the uncertainty due to variations in reported refractive indices (Laskina et al., 2014; Johnson et al., 2020; Myers et al., 2020). The variation of the slope from the $\alpha_{10}$ (for homogeneous medium) is consistent (+20%) for both the peaks. The higher slope of the peak near 1110 cm$^{-1}$ (as also evident

from its higher values in Figure 5a) is not because of Si-O-Si band (1130 - 1100 cm$^{-1}$) from the silicone grease, removing which the slope still varies by 18% (Section S3 Figure S6b). In our experiments the peak height ratio of $\nu_3(SO_4^{2-})$ to $\nu_4(NH_4^-)$ was 2.1 ($\pm10\%$); consistent with spectra acquired from nebulized and dried polydisperse ammonium sulphate solution (4.8 g/L) with mass mean diameter of 200 nm studied under conditions close to our experiments (Weis and Ewing, 1996). The consistency of peak height ratios and similarly higher slopes from the linear absorption coefficient (for homogeneous medium)

further strengths the observation that the particles deposited on the crystal behave more like a homogeneous medium.

In the analysis above, the ZnSe crystal was not assumed to play a role. Nonmetallic particles collected on the surface of a substrate can behave optically different from suspended particles due to multiple far-field scattering interactions, van der Waals-like interactions, and superposition of incident and reflected electromagnetic fields between particle and surface (Quinten, 2011). Scattering power of IR radiation by submicron particles is reasonably small (though increasingly important in the region

above 1500 cm$^{-1}$ for larger particles within this range), so these far-field (incoherent) interactions are not considered to be substantial. Otherwise, deviations in our measurements were within uncertainty of past reference measurements, and systematic influences of near-field optical effects or physical interactions were not detected within the precision of our technique.

We also evaluate the value of optical microscopy experiments (and corresponding image analysis) in our quantification by considering a case where a constant density scaling factor is employed — i.e. $\phi_R(r_b) = 1$ for all samples (Section S3 Figure

S6a), which results in a shift of the points along the abscissa for each experiment compared to Figure 5b. The correlation with absorbance remains almost the same ($R^2$ of 0.96 and 0.95) for the 1110 and 1410 cm$^{-1}$, respectively. The estimated slope also remains statistically indifferent, at $1.3\times10^4$ cm$^{-1}$ (95% CI of $[1.14, 1.51]\times10^4$ cm$^{-1}$) for the peak near 1110 cm$^{-1}$ and $5.9\times10^3$ cm$^{-1}$, (95% CI of $[4.95, 6.84]\times10^3$ cm$^{-1}$) for the peak near 1410 cm$^{-1}$. The results of the analysis are effectively the same regardless of whether image analysis is incorporated, because $\phi_R(r_b)$ was effectively close to unity (Section S3 Figure S7b).

Finally, we estimate a limit of qualification (LOQ) of $m_a^*(r_b) = 0.012$ µg/cm$^2$ for ammonium sulphate based on the threshold at which the absorption at 1410 cm$^{-1}$ is ten-fold greater than root mean square of noise (Section S6). This threshold is at least an order of magnitude lower than for PTFE filters based on reported accounts for other substances or peaks (Russell et al., 2009; Debus et al., 2021), which is an expected result due to low substrate interferences and baseline uncertainties. For our current ESP device operating at a flowrate of 2.1 LPM with overall efficiency of 0.64 (for charger and collector), this LOQ



translates into a required average airborne concentration threshold of 1.5 µg/m$^3$ of ammonium sulphate over a sampling time of 30 minutes. PTFE sampling can potentially achieve similar time resolutions for quantification of airborne concentrations due to high particle collection efficiency and flow rate (at the cost of greater pressure drop) than our ESP, which is limited by the applicable voltage to maintain the strength of electrostatic forces that facilitate particle deposition. However, greater sensitivity to deposited mass on the optical crystal through ESP collection may confer benefits when the flow rate or total sampled volume

is limited, such as in cases of co-sampling from manifolds, environmental chambers, flow reactors, or denuders. Furthermore, the lower pressure drop and face velocity in the ESP is expected to reduce evaporation artifacts from semivolatile substances reported for filter sampling (McDow and Huntzicker, 1990; Zhang and McMurry, 1991).

## 4   Conclusions

This work demonstrates the possibility of using a 3D printed radial ESP for quantitative IR spectroscopy with minimal optical

interferences. Rapid prototyping to achieve the presented design was achieved through numerical simulations and 3D-printing with ABS. To enable 3D-printing of parts for a flow system, the number of joints were minimized by printing with a 45° rotation to allow fabrication of parts with overhangs, and a small amount of acetone was used for press-sealing surfaces after printing. Polydisperse ammonium sulfate particles were atomized from solution and studied over 11 experiments with varying particle loading. The collection area and spatial distribution of particles on the crystal were analyzed using optical image analysis

and IR absorbance measured across variable aperture sizes. The mass distribution was estimated to be semiuniform due to the variation in electrode distances and electric field strengths within the collector. We observed the IR absorbance at $\nu_3(\mathrm{SO_4^{2-}})$ and $\nu_4(\mathrm{NH_4^+})$ peaks at $\sim$1410 and $\sim$1110 cm$^{-1}$ to scale linearly ($R^2 > 0.94$) with particle loading over two orders of magnitude, with an estimated limit of quantification of 61 ng of collected mass, corresponding to an average airborne concentration of 1.5 µg/m$^3$ for a 30 minute sampling interval. The peak positions and peak heights of the extinction spectra were within the

variability of previous studies using particles suspended in an inert gas, and particle-substrate effects were not discernable. While particle extinction should follow the electrostatic approximation for small, separated particles (for areal mass densities $< 3$ µg/cm$^2$ and size $< 1$ µm in this work), we find that modeling the particle deposits formed in this work as homogeneous medium of equivalent mass also provided a reasonable prediction of the apparent absorbance. The advances presented in this work permit higher sensitivity and chemical resolution for aerosol measurement, and encourage 1) further investigation of this

design and analysis strategy for collector design, and 2) use of such a device in studies of aerosol composition for various applications.





**Figures**

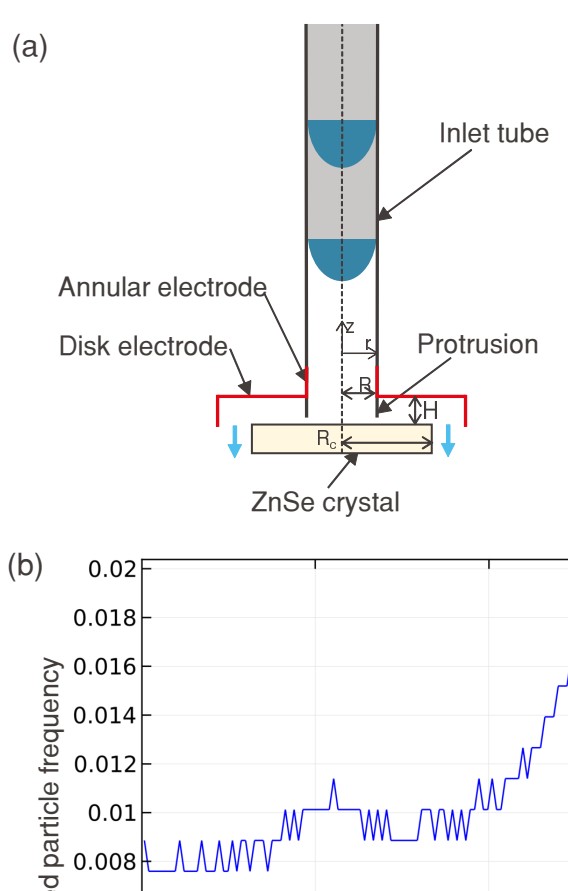

**Figure 1.** (a) A schematic representation of the radial ESP where particles enter through an inlet tube that is extended closer to the collection crystal through a protrusion. The particles are focused on the crystal using high voltage electrodes near the exit of the tube, and above and around the crystal. (b) Radial particle deposition profile as obtained from COMSOL simulation of velocity, electrostatics and particle tracing on the device in part a - the mound of particles near $r = 5$ mm corresponds to the position of the protrusion.



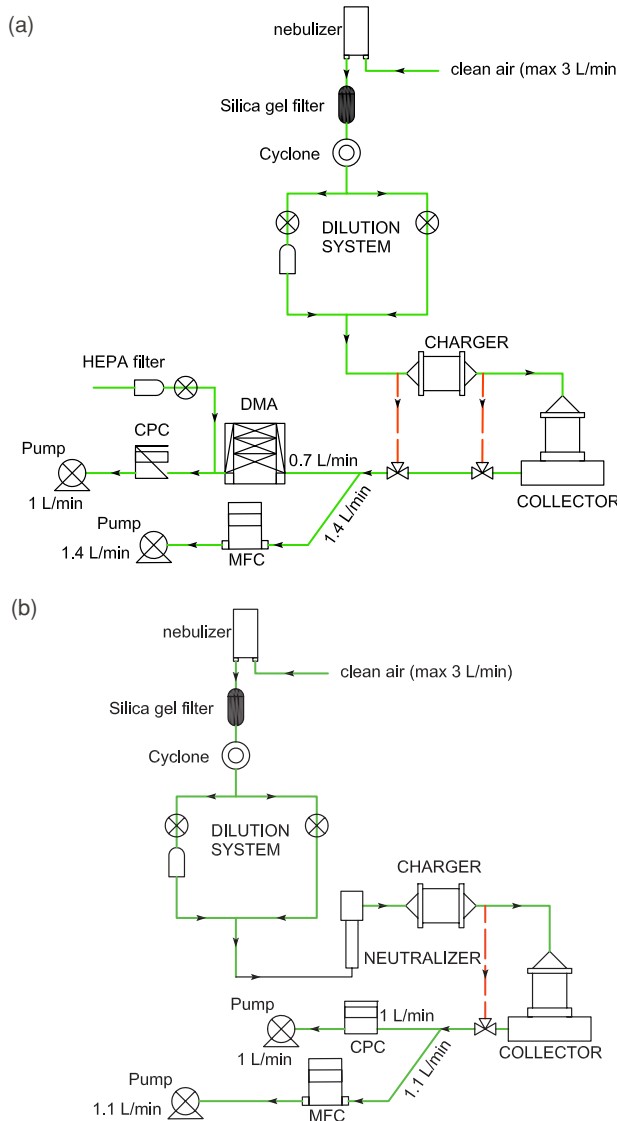

**Figure 2.** Diagram of the aerosol flow system experiments (green line is the main flow line and the red-dotted lines are the alternate bypass lines) for (a) obtaining the particle size distribution, and (b) obtaining the mass loading reference through the CPC particle counts.



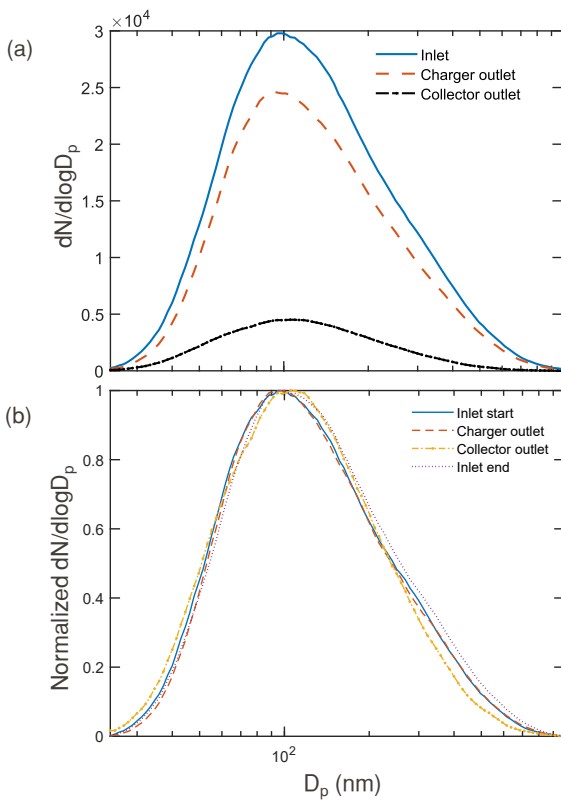

**Figure 3.** (a) Particle size distribution measured at the inlet averaged from that before and after collection, at the charger outlet averaged from that before and after collection and at the outlet of the collector, and (b) normalized particle size distribution measured for the inlet before staring the experiment, at the outlet of the charger, at the outlet of the collector, and the inlet at the end of the experiment (to determine any bias in the inlet).



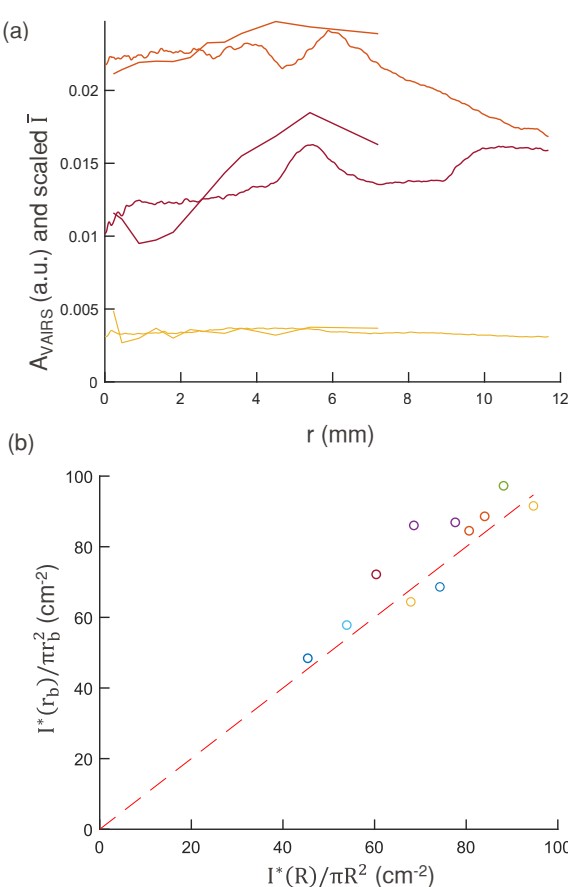

**Figure 4.** (a) Radial profile of scattered intensites ($\bar{I}(r)$), obtained from image analysis (Section S5) and from VAIRS analysis ($A_{\text{VAIRS}}$). $\bar{I}(r)$ values range to a higher $r$ and have higher resolution than $A_{\text{VAIRS}}$ values and are scaled differently for each sample to display it together with the corresponding $A_{\text{VAIRS}}$ profile. (b) Calculated $I^*(r_b)/\pi r_b^2$ and $I^*(R)/\pi R^2$ from image analysis for the experiments with different $m_a^*(R)$ and a 1:1 reference line (red-dotted line).

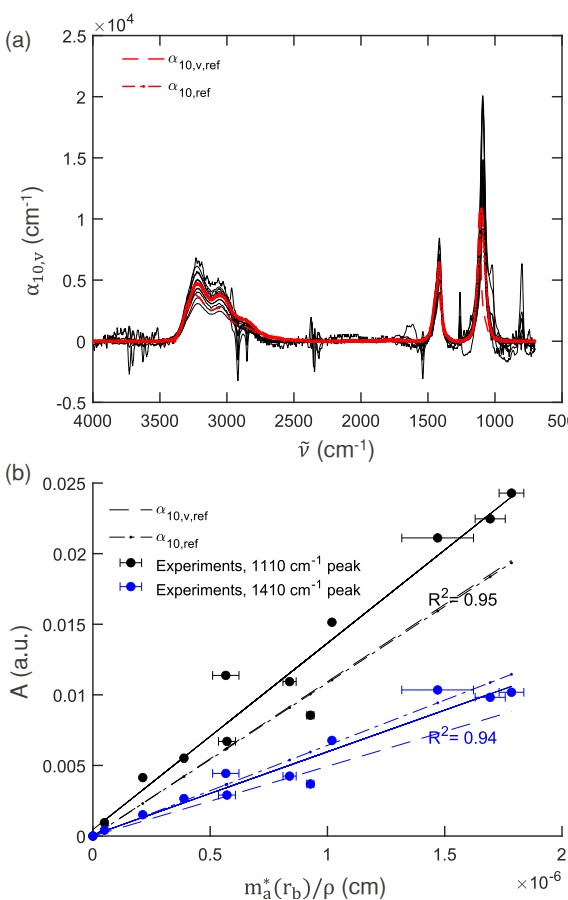

**Figure 5.** (a) Comparison of volume absorption coefficient ($\alpha_v$) from the experiments for IR measurement with an aperture of 6mm against the reference linear absorption coefficient ($\alpha_{ref}$) and volume absorption coefficient ($\alpha_{v,ref}$) for ammonium sulfate calculated using $n$ and $k$ (Earle et al., 2006). (b) Response of IR absorbance ($A$) against the effective deposition thickness (cm) of the total particle collected reference from CPC calculated using image analysis, for absorbance at the peak near 1110 cm$^{-1}$ (for $\nu_3(SO_4^{2-})$) and at the peak near 1410 cm$^{-1}$ (for $\nu_4(NH_4^-)$).



*Author contributions.* ND designed and fabricated the device, ran the simulations, prepared and performed the experiments, and performed the data analysis. ST conceived of, acquired funding for, and provided supervision of the project, and provided feedback on the analysis. ND
and ST wrote the manuscript.

*Competing interests.* We declare that no competing interests are present.

*Acknowledgements.* We thank Dr. Jiannong Fang for discussions on numerical simulations and SNF 200021_172923 for funding.



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
