# Peer review of "Design and fabrication of an electrostatic precipitator for infrared spectroscopy"

_Atmospheric Measurement Techniques, 2022_

## Author Response (AR2)

**"Design and fabrication of an electrostatic precipitator for infrared spectroscopy" - response to reviewer comments**

Nikunj Dudani and Satoshi Takahama

**Reviewer 1**

I have no general nor technical comments. The methodology is promising and I hope the authors will venture in utilizing the methods to see how results will vary if we have to consider other aerosols for the study, as mentioned on their future study priority 2.

We thank the reviewer for a review, corrections, and encouraging remark.

Minor comments/changes as follows:

- Line 67: recommend adding a statement why ABS was used for 3D printing.

  The intercomparison of ABS with the other contending materials (Nylon and PLA) and the factors leading to the decision is specified in the first paragraph of section 2.3:

  "We discuss three considerations for fabrication: material selection, printing protocol, and post-printing assembly and treatment. For rapid-protyping there are limited materials that can be 3D-printed reliably: nylon, ABS and polylactic acid (PLA); these materials have tradeoffs in print reliability, durability, chemical interferences, and electrical properties. PLA is not stable to heat, less durable, and has known outgassing issues, limiting its application for aerosol chemical analysis. Nylon is a stronger material than ABS but lies higher in the triboelectric series, making it prone to more electrostatic losses of particles near the surface. Moreover, in our observations the main source of leaks in the initial prototypes was through the in-layer-space between each print layer. The spacing was much higher in nylon as it can absorb moisture during and after printing and results in larger printing defects and layer-separation. The additional strength of nylon comes with cost of being more brittle and harder to seal, leaving ABS as our choice of material for this application. The strength of ABS is sufficient for inherently low pressure drop ESP applications as it has a yield strength of $\sim$25 MPa, which is much higher than the expected hoop stress of $\sim$1.7 MPa acting on a 50 mm internal diameter cylinder design with a thin 2 mm ABS body operated with an extreme 1 atm pressure."

- Line 94: add the word "the" after the word making

  We edited this in the revised text.

- Line 138: Is "satisficed" the correct term or are you referring to "satisfied"?

  We have used "satisficed" to emphasize that we halted the development beyond a certain minimum level envisioned for the quantitative performance evaluation.

- Line 201: recommend adding a statement/s on why ammonium sulfate was selected as an ideal compound for the study.

  Included the following explanation in the revised text: "...mainly because the solid near-spherical particles generated are non-volatile and allows us to neglect any mass change between collection and IR measurement."

- Line 384: Are we missing some information after refractive indices?

  We edited the sentence in the revised text to add clarity:

  "The ratio of $\alpha_{10,v}$ to $\alpha_{10}$ deviates from unity — as calculated from reference spectra for ammonium sulphate (**?**) (Section S3 Figure S5a) — especially near the absorption peaks."

**Reviewer 2**

The manuscript describes a method for direct collection of submicron particles onto a circular ZnSe window using a custom designed radial electrostatic precipitation device. This appears to be a promising method that overcomes a number of disadvantages involved with other methods of particle collection and allows functional group analysis with transmission FTIR spectroscopy.

There are many grammar errors in the manuscript that added to the review time and prevented my understanding in several places. In addition, there are many typos that also added to the review time. It became clear throughout the manuscript that the authors did not proofread their work, including their supplemental material. I do believe the method is interesting and the effort on the analysis is useful.

We thank the reviewer for a thorough review, corrections, and encouraging remark.

Overall questions and comments:

1. Line 34-35 – You may also want to cite the following more recent and relevant papers that use ATR crystals for impaction substrates:

   Kidd et al., PNAS, 111, 2014, "Integrating phase and composition of secondary organic aerosol from the ozonolysis of a-pinene", doi: 10.1073/pnas.1322558111

   Yu et al., J. Environ. Sci., 7, 2018, "Fast screening compositions of PM2.5 by ATR-FTIR: Comparison with results from IC and OC/EC analyzers", doi: 10.1016/j.jes.2017.11.021

   We thank the reviewer for the suggestion. We are aware of these and other works that have reported aerosol impaction followed by ATR-FTIR analysis, but had intended to highlight earlier examples. We have now cited them in our revised text.

2. Line 135 – Does this sentence mean every diameter between 100 and 800 had 75% collection efficiency? The mention of 200 nm and then a listing of other diameters is confusing.

   We modified the sentence in the revised text to clarify the range of efficiencies for the simulated diameters. "A simulation of the device with inlet radius of 10 mm and flow rate of 2.1 LPM, $E_0 = 1$ kV/mm, and 1 elementary charge per every 20 nm diameter for 100 nm $< D_p <$ 800 nm, resulted in a mean collection efficiencies between 65% and 90% (mean of 75%), with smaller particles showing higher efficiency. An overall increase in collection efficiency of 5% (mean of 80%) was achieved in the simulations by incorporating a protrusion that extended the inlet tube closer to the collection surface while keeping the electrodes at a farther distance (Section S1 Figure S1a), a feature we incorporated in the final design."

3. Line 326 – Can the authors clarify where the number of elementary charges come from? Are these values measured? They state they had approximately 1 charge per every 20 nm, but it looks like it has a trend.

   The number of elementary charges are not explicitly measured. We have rephrased the sentence in the revised text to "From the numerical simulations, the charge level that would result in collection efficiencies for all sized around the observed 82% is observed to be between 1 elementary charge every 25 nm diameter for the smaller particles and 1 elementary charge every 17 nm diameter for larger particles. The variation in the charge level is not implausible

and could have contributed in part to the observed higher efficiency for the larger particles, though a thorough analysis of the charger is planned to estimate the exact charge values."

4. Line 325 – the authors say 75% collection efficiency for these particles earlier in the text and then 82% in another place. Can you clarify?

The revisions made to the text in the points 2 and 3 also clarifies on the difference between the two efficiency values. 75% efficiency was obtained in the numerical simulations without the tube protrusion. By including the protrusion, the efficiency was higher (80%) in the simulations. The $82\% \pm 8\%$ efficiency mentioned here are the observed values estimated from the particle size distribution measurements. This has been clarified by editing the sentences, as mentioned in points 2 and 3 above.

5. Line 333 – the optical images are in Figure S3, not Figure 4a.

"The radial dependence of pixel intensity" has been shown in Figure 4a and not the optical images itself. Changed the position of "(Figure 4a)" reference to be right after "pixel intensity" in the revised text.

6. Line 374-382 – I'm not sure I understand why the resolution of an older dispersive instrument would matter. If you are concerned that the peak position measurements were made with a dispersive instrument but yours were made with an FTIR, can you not compare the peak positions of (NH4)2SO4 in newer references using FTIR? In addition, measuring your own reference peak positions by putting (NH4)2SO4 on the surface in a different manner would help. If the description on line 400 is similar to this suggestion, maybe you can begin that section by explaining there was a comparison made to homogeneously deposited (NH4)2SO4 to show that it is consistent before you present your analysis. However, I'm not sure I understand what you mean by homogeneous medium.

The refractive indices reported using the dispersive instrument has a wavenumber resolution of 20-30 cm$^{-1}$ in this region, preventing determination of peak position with the precision of newer FTIR measurements with wavenumber resolution reported at 2-4 cm$^{-1}$ (including ours, and that of Earle et al. 2006 used for comparison in our work). In identifying the maximum absorption at the peak, we therefore use the refractive indices of Earle et al. We modify the text as follows:

"[...]the lack of wavenumber resolution in the older measurements made on dispersive spectrometers prevents a more precise comparison of the peak position (20–30 cm$^{-1}$ rather than 2–4 cm$^{-1}$ made with Fourier transform instruments for condensed-phase samples)."

By "homogeneous medium", we refer to the optical equivalent of a solid crystal of ammonium sulfate that does not include effects of the crystal surface (i.e., dielectric boundaries that gives rise to reflection and scattering). Johnson and co-workers (https://doi.org/10.1177/0003702820930009doi:10.1177/0003702820930009 and https://doi.org/10.1177/0003702820928358doi:10.1177/0003702820928358 and recently report similarities in refractive indices of powders pressed into a compact pellet resembling a homogeneous medium, but that is not the intended implication of our work. For the purpose of this study, we include the calculation of the linear absorption coefficient that would correspond to this homogeneous medium, since this assumption is often implicitly invoked in chemical spectroscopy (regardless of sample morphology). We did not intend to imply that the particle ensemble is behaving as a homogeneous medium, but that modeling them as such do not introduce large errors.

We modify the text as:

"Overall, the uncertainty in the measurements enveloped the difference in modeled optical

response of ammonium sulphate as an ensemble of particles or (hypothetical) surface-free homogeneous medium of equivalent thickness."

Minor comments:

- Use electrostatic precipitator as a keyword, spelled out in addition to abbreviated.

  We added "electrostatic precipitation" in the list of keywords in the revised text.

- Line 7 – extra word "aerosol" here?

  We removed the word in the revised text.

- Line 36 – spell out ESP the first time it is used.

  We added the "ESP" acronym in line 7 where it was mentioned for the first time.

- Line 60 – your reference formatting needs some attention. These should be in parentheses.

  We changed the references accordingly.

- Line 84 – you need to define PEBS. Change this to "the personal electrostatic bioaerosol sampler/ing (PEBS) and the PEBS's wire-to-wire electrode ..."

  We modified the sentence accordingly.

- Line 87 – What does factor of safety 3 mean? Do you mean "three times over the theoretical breakdown?

  We modified the sentence in the revised text to "..a factor of three lower than the theoretical breakdown" and removed the mention of "Factor or safety (FoS)" used more commonly in mechanical and electrical system design.

- Line 94 - Rewrite this sentence; grammar issue; it is not clear what you are describing here.

  We modified the sentence in the revised text to "For a given flow condition, a stronger electric field will result in a higher particle collection efficiency, resulting in the closest separation distance of 5 mm being the most desirable, as any larger separation distance would either require that $V_0$ be higher or $E_0$ be lower than 1 kV/mm."

- Line 96 – The supplemental figures are out of sequence. They should be numbered in the same order as they are described in the text.

  We rearranged the figures in according to the order in the main text.

- Line 117 – what is meant by a scrupulous flow field?

  We change to "erroneous flow field" in the revised text.

- Line 125 – the correct terminology is slip correction factor.

  We corrected in the revised text.

- Line 126 – you may also want to see the book "Aerosol Technology: Properties, Behavior, and Measurement of Airborne Particles" by W. C. Hinds, 1999 for information relevant to this work.

  This has now been included as a reference.

- Line 135 – Line 122 states that you believe there is one charge for every 20 nm, but the 200 nm particles were estimated to have 8 charges instead of 10. How many charges were on each of the other diameters?

The sentence has been modified to clarify the charge levels, as specified in the point 2 in the "Overall questions and comments" section.

- Line 348-349 – grammar issues in this sentence, not sure what this is trying to say.

  We modified the sentence in the revised text to "The estimated density scaling factor $\phi_R(r_{\rm b})$ is the ratio of $\left(I^*(r_{\rm b})/\pi r_b^2\right)$ to $\left(I^*(R)/\pi R^2\right)$ (the two axes in Figure 4b for a beam radius $r_{\rm b} = 5.4$ mm — corresponding to the aperture diameter of 6 mm). The deviation of $\phi_R(r_{\rm b})$ from unity (corresponding to the 1:1 line in the Figure) is mostly $\pm 10\%$ (with two points at $+20\%$ and $+25\%$), with its value being systematically higher for the intensity profiles that increases sharply with radius and systematically lower for profiles which have a decreasing radial intensity (Figure 4a)."

- Line 384, 396 – there are several places where NH4- has been used instead of NH4+ including in the supplemental and figures!

  Corrected in the revised text.

- Line 391 - I don't understand this format shown in several places. What is 1.11, 1.52? Is there something missing?

[revised manuscript text omitted]
 physical device (Figure S1) and numerical simulations (Figure S2) are shown below. The numerical simulations in COM-SOL Multiphysics was performed on a class of devices illustrated in the schematic of Figure 1a to obtain the steady state velocity and electrostatic field, to use for the transient particle tracing simulation and obtain the radial particle distribution on the crystal (Figure 1b). The device geometry from the simulations was replicated using 3D-printing a few parts  that allow removing and replacing the crystal from the device.

[Figure]

**Figure S1.** (a) Engineering drawing of the device assembly, (b) cross-section view with dimensions (in mm),(c) side view of the electrode housing part with the protrusion visible, and (d) fabricated and assembled device with the different parts labelled.

[Figure]

**Figure S2.**  Numerical simulations of the  ESP device for (a)  the  device schematic with the outer and inner body (ABS)  highlighted, b)  steady state voltage field (e), c)  steady state velocity field (m/s), and d)  particle trajectories in the steady state field at the last time step.

**S2  Baseline correction**

The spectra are baseline corrected using a smoothing spline fitting and subtraction (Kuzmiakova et al., 2016). The background points to which the smoothing spline is fitted can be identified using a number of methods (e.g., derivatives, mixture models, and asymmetric weights) (Liland et al., 2010; de Rooi and Eilers, 2012). In this work, background points were identified by setting limits on the moving mean and moving standard deviation on the slope of absorbance spectra. As the change in slope for sharp peaks or even broader peaks is higher than that for the baseline regions, using a combination of the change in the mean value and the deviation in slope allowed separating the baseline form the absorbance regions. For identifying the broad peaks (e.g. 2700 cm$^{-1}$ - 3200 cm$^{-1}$), the moving mean of the absorbance itself is additionally used to separate it from the background.

[Figure]

**Figure S3.** (a) Absorbance spectra (blue), moving mean of the absorbance spectra (red), slope of the absorbance spectra (orange), moving mean of the slope (green) and te moving standard deviation of the slope (purple). (b) Identified baseline points (green points). (c) A smoothed spline fitting the baseline points (green), and (d) the baseline corrected spectra.

**S3 Quantitative analysis**

The particle deposit was qualitatively analyzed using scanning electron microscopy (SEM) (Figure S5a, b, c) and optical microscopy (Figure S5d) to obtain the average radial scatter (Figure S5e).

[revised manuscript text omitted]